# Effect of health education intervention on treatment adherence and health status in patients with Chronic obstruction pulmonary disease: A random control trial

Nguyen Thi Thu Trieu[1☉], Nguyen Thi Yen Hoai[1☉], Nguyen Ngoc Tung[2‡],
Tran Huu Thong [3¤a,4¤b,5¤b‡]*, Nguyen Thi Thu Hien[6¤a,7¤b‡]

**1** Faculty of Nursing, Da Nang University of Medical Technology and Pharmacy, Da Nang city, Vietnam,
**2** Can Tho University of Medicine and Pharmacy, Can Tho city, Vietnam, **3** Center for Emergency
Medicine, Bach Mai Hospital, Hanoi, Vietnam, **4** Department of Emergency and Critical Care Medicine,
Hanoi Medical University, Hanoi, Vietnam, **5** Faculty of Medicine, University of Medicine and Pharmacy,
Vietnam National University, Hanoi, Vietnam, **6** Dermatology & Burn Department, Bach Mai Hospital,
Hanoi, Vietnam, **7** Department of Nursing, Bach Mai Hospital, Hanoi, Vietnam

☉ These authors contributed equally to this work
‡ These authors also contributed equally to this work.
¤a First current address
¤b Second current address
* thongccbm@gmail.com

org/10.1371/journal.pone.0325192

Medical Center, UNITED STATES OF AMERICA

**Peer Review History:** PLOS recognizes the
benefits of transparency in the peer review
process; therefore, we enable the publication
of all of the content of peer review and
author responses alongside final, published
articles. The editorial history of this article is
available here: https://doi.org/10.1371/journal.
pone.0325192

## Abstract

Adherence to treatment is critical to effective management of COPD and is key to
addressing the growing burden of disease. So, this study conducted to evaluate the
effectiveness of a health education intervention on treatment adherence behavior
of COPD outpatients and identify the level of health status improvement. A random
control trial was conducted in 2022 at two respiratory outpatient clinics in Da Nang
City, Vietnam. 90 participants were divided into two group, 45 members of the health
education group who received 5-times consulting about disease knowledge, train-
ing using inhalers, breathing exercises at clinic, then 12-times teleconsultation;
and 45 controls who only joined surveys. Treatment adherence and level of health
status improvement were assessed as outcomes. At beginning, there was no signif-
icant difference between the intervention and control groups regarding compliance,
disease severity, and airway obstruction ($p > 0.05$). After three months of interven-
tion, adherence to inhaled medication in the intervention group was 3.2 times higher
than in the control group, with statistical significance ($p = 0.002$, OR=3.2; 95% CI:
1.8–5.3). Similarly, adherence to breathing exercises was 2.5 times greater in the
intervention group compared to the control group, with a statistically significant result
($p = 0.000$, OR=2.5; 95% CI: 1.8–3.4). Overall adherence to treatment was 2.2 times
higher in the intervention group than in the control group, also showing statistical
significance ($p = 0.001$, OR=2.2; 95% CI: 1.7–3.1). Additionally, the intervention group
demonstrated significant improvements in disease severity and airway obstruction

**Data availability statement:** The datasets are available in Figshare. (DOI:10.6084/m9.figshare.28684955)

**Funding:** The author(s) received no specific funding for this work.

**Competing interests:** The authors have declared that no competing interests exist.

(p < 0.001). Education interventions showed effectiveness in increasing treatment adherence and health status of COPD outpatients. Therefore, a broader program should be conducted in the future.

## Introduction

Chronic obstructive pulmonary disease (COPD) is the third leading cause of death and accounted for 6% of all deaths in 2019 in the world, of which more than 80% occurred in low- and middle-income countries. In addition to increased morbidity and high mortality, COPD also causes a significant socioeconomic burden in low- and middle-income countries due to its impact on work productivity [1–2].

Treatment adherence is critical to effectively managing COPD and is vital to addressing the growing burden of disease.[3] Treatment adherence is often assessed through medication adherence and breathing exercise adherence. Recent studies have found that regular, correct dose, correct timing of inhaled medication, and taking medication even when there are no symptoms of the disease are essential to improve lung function and increase exercise capacity and reduce the frequency of exacerbations [3]. Meanwhile, regular implementation of breathing exercises can lead to acute improvements in gas exchange and ventilation, reduced dyspnea, exercise capacity, and health-related quality of life for COPD patients [4]. Compared with other treatments, breathing exercises are easy to operate, have no location restrictions, and do not require much funding, which can significantly improve patient initiative and compliance [3].

However, many studies have shown that long-term adherence to chronic disease treatment is low, and it is estimated that only 50% of responding patients comply with treatment [5]. Missing doses and stopping treatment early, fear of side effects, or becoming dependent on the medication over a long period of time are common reasons why most patients are non-adherent to medication [6]. Although there is strong evidence of the beneficial effects of breathing exercises, in reality, adherence to breathing exercises in COPD patients is rarely practiced regularly in daily life. The most important barrier to adherence to breathing exercises is the fear of difficulty breathing, which can lead to a lack of intention to practice and thus reduce breathing practice because patients worry about the feeling of difficulty breathing that they cannot control when practicing breathing exercises [7]. Assessing treatment adherence is a challenge in the clinical assessment of patients and in science research.

Many intervention studies have been carried out to promote treatment adherence for COPD patients worldwide, including in Vietnam, but they were conducted alone or only focused on improving medication adherence or building rehabilitation programs through guided breathing exercises [8–10]. Specifically, the intervention by Khadela et al. (2020), and Ibrahim and El-Maksoud (2021) only focused on improving knowledge and self-management of the disease [11]; while Yang et al. (2022) reviewed studies that improved adherence to breathing exercises rather than medication adherence [12]. In Vietnam, Tu et al. (2019) showed that after the intervention, patient adherence increased from 37.4% to 53.2%, however, the intervention had no

control group, was implemented by pharmacists, so it only focused on medication adherence and only evaluated effectiveness by symptom scores with low validity [13].

According to the GOLD 2024 guidelines, treatment adherence for individuals with COPD encompasses more than just medication compliance. It requires a comprehensive approach that includes various measures such as avoiding risk factors, quitting smoking, and undergoing respiratory rehabilitation [14]. Patients must strictly adhere to all these strategies to slow the progression of COPD and prevent severe complications.The GOLD 2024 guidelines emphasize the critical role of adherence in managing respiratory conditions. Interventions are designed to include patient education sessions that promote proper inhaler use, effective breathing exercises, and compliance with prescribed therapies. This approach prioritizes ongoing education and regular follow-up visits, aligning with GOLD's focus on empowering patients through continuous education and comprehensive care [3].

Intervention research focusing on promoting treatment adherence, including adherence to inhaled medications and adherence to respiratory rehabilitation through breathing exercises in Vietnam, has not been found in any recent publication. Therefore, we conducted the study to evaluate the effectiveness of the health education intervention with the aims of evaluating the effectiveness of a health education intervention on treatment adherence behavior of COPD outpatients and identifying the level of health status improvement of those patient after receiving the intervention.

## Materials and methods

### Study design and Participants

A random control trial was registered in Thai Clinical Trials Registry with the identification number TCTR20240526001. The study conducted in accordance with CONSORT statement (https://www.equator-network.org/reporting-guidelines/consort/) from March to December 2022 "S1 File". The study protocol was published on https://www.protocols.io/view/study-protocol-d6269ghe [PROTOCOL https://doi.org/10.17504/protocols.io.6qpvrko8blmk/v1] "S2 File".

The study participants were taken from respiratory outpatient clinics at C-Da Nang Hospital, and Da Nang Lung Hospital, Vietnam. Both hospitals share comparable features regarding facilities, quality of care, and patient demographics. Selection criteria include patients who were: 1) treated with inhaled medications at home in the stable phase and do not have an exacerbation requiring hospitalization for at least three months; 2) can speak, read, and understand Vietnamese; 3) owned and used a mobile phone frequently; and voluntarily agreed to participate in the study. Partcipants were excluded if they: 1) had a history of bronchial asthma, allergic rhinitis, lung surgery, or other respiratory diseases; 2) were experiencing a COPD exacerbation or exacerbation of co-morbidities in the past three months or have had a change in medication in the past three months; 3) or people with mental disorders or other serious illnesses.

### Sample size

The sample size was calculated using the sample size formula for comparing two proportions. Accordingly, the estimated treatment adherence rate before intervention was 50%, and the expected treatment adherence rate after the intervention was 85%, with values of $\alpha = 0.05$ and $\beta = 0.1$ [13]. In addition, the attrition rate may occur at 30% with study subjects being absent and/or not cooperating in the study continuously for the entire 3 months, it was estimated that each group needed to have 45 participants.

### Randomization and blinding

Patients for the study were selected based on predefined inclusion and exclusion criteria. A simple random sampling method was used to allocate participants into the control and intervention groups. At the beginning of the study, patients were randomly assigned to either the intervention or control group using a block randomization method. This approach was chosen to ensure similarities in treatment adherence, degree of airway obstruction, and disease severity, which were

deemed essential factors in the home management of COPD [15,16]. The block randomization was generated by a computer, facilitating an orderly distribution into two groups and minimizing the risk of uneven allocation.

In terms of stratification, both groups showed similarities in treatment adherence, degree of airway obstruction, and disease severity. A simple random sampling approach was then applied to assign individuals from these stratified groups to either the intervention or control group until the target sample size of 45 participants in each group was reached. Recruitment was carried out through telephone calls, with the research support team contacting potential participants to invite them to join the study based on the inclusion and exclusion criteria. Calls continued until the desired sample size was achieved. All participant information was systematically recorded in a computerized data file.

It is important to note that we were unable to implement a blinded study, which is often challenging in trials assessing non-pharmacological interventions such as rehabilitation or behavioral therapies [17,18]. Both participants and researchers were actively involved in the health education intervention, meaning both were aware of the treatment being received. Additionally, direct interactions occurred between patients and investigators during clinical examinations, counseling, and assessments.

## Intervention

The intervention conducted from 01/03/2022–25/12/2022.

Participants in the intervention group participated in two topical discussions with the research team at the outpatient clinic. Within 60–90 minutes, patients were provided with knowledge about the disease, practical skills in using inhaled drugs, instructions on breathing exercises, and self-management skills to improve their treatment adherence. The second session was conducted one week later "Table 1". In addition, after completion, each patient would be given and instructed to maintain medicine use and breathing exercise diary. The diary was collected after the intervention ended and considered the basis for assessing participants' treatment adherence level. Face-to-face meetings were still conducted once a month for three continuously 3 months when the patient came for a regular appointment. This outline program has assessed by healthcare managers and practicing nurses with high scores of acceptability, appropriateness, and feasibility (M = 4.31; SD = 0.11) and (M = 4.37; SD = 0.12), respectively [19].

The online home monitoring process was conducted immediately afterward and continued for 3 months. Periodically once a week during the hours of 8–9 am on Wednesday of the week, the research team made group phone calls (5 patients/group) via Zalo software; the time for each phone call was around 3–5 minutes and no more than 10 minutes. For

**Table 1. Outline of educational intervention program topics.**

| Topics and components in the theory | Intervention contents |
|---|---|
| **Topic 1:** Focus on improving knowledge about COPD<br>**Theoretical component:** Attitude towards behavior and subjective norm | - Determine the patient's level of knowledge<br>- Determine the patient's attitude in receiving knowledge related to the disease<br>- Determine the level of knowledge the person needs to receive support from the nurse<br>- Analyze for patients to understand the benefits of understanding the disease in disease prevention and control. |
| **Topic 2:** Guidance on self-management methods<br>**Theoretical component:** Perceived behavioral control | - Discuss self-management measures<br>- Discuss the advantages and disadvantages of not practicing self-management behaviors<br>- Determine the ability to self-implement the self-management disease methods<br>- Discuss the difficulties of the patient when self-administering the self-management disease methods |
| **Topic 3:**<br>- Increase the rate of correct practice of using inhaled medication<br>- Guidelines for the COPD self-management action plan<br>**Theory component:** Improved behavior | - Determine the perceived level of control over the use of inhaled medications<br>- Instructing participants to strictly follow the standard steps in drug use<br>- Encourage the participants to do the right thing, emphasizing the steps to perform the inhaler technique that often make mistakes or missed, increasing level of behavioral control as well as possible<br>- Provide video instructions on how to use inhaled medicine for patients to watch when needed<br>- Guild patients writing medical consumption diary<br>- Encourage patients to make their own action plan to adhere to treatment |

patients who did not participate in the group call, the researcher called via their personal phone number to remind them to participate. The private call would be made three times, each time five minutes apart, to ensure the group call had enough participants. Participants were instructed not to tell and/or share phone calls contents with others. During these calls, each participant self-reported medication history, side effects of the drug (if any), the process of performing breathing exercises at home, common symptoms when performing therapy, number of dyspnea/weeks, amount and color of sputum. In addition, the research team also provided some health information such as measures to deal with dyspnea, reminding patients to practice these exercises, advising to quit smoking, practice inhalation with Sopiroball, practice coughing effectively and reminding patients to record information in the diary, send a video of their breathing exercises for us to monitor and support. Throughout all group phone calls, we maintained a consistent structure by implementing these discussed strategies to ensure uniformity in our communication "Fig 1".

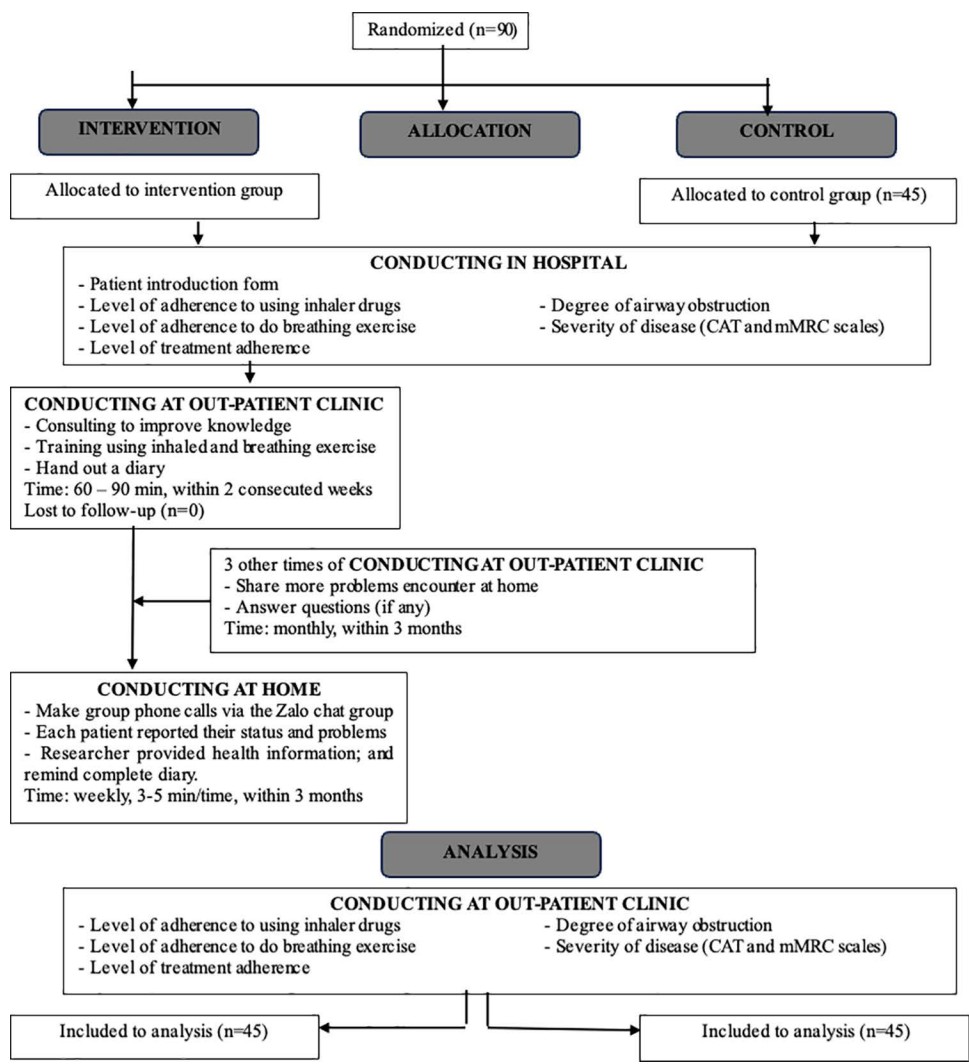

**Fig 1. Study design.**

## Data collection

The primary outcome in this study was treatment adherence (including adherence to inhaled medications and adherence to breathing exercises), and the secondary outcome was the proportion of participants with health status enhancement (assessing by disease severity, degree of dyspnea, and degree of airway obstruction). Primary and secondary outcomes were measured at study baseline and three months later.

Treatment adherence was assessed in two ways, including adherent to inhaled drugs, and adherent to breathing exercises. Assessment of adherence to inhaled drugs by the Test of Adherence to Inhalers (TAI-10) that developed by Plaza et al., consisted of 10 questions [20]. Each item is based on a 5-Likert scale that ranges from 1- worst to 5 – best adherence. The total score was from 10 to 50 points, in which patients were seen as adhering with a score ranging from 46 to 50, and non-adherence for a score ≤ 45. The questionnaire was testing reliability with high score (Cronbach alpha at 0.871), and the test-retest reliability coefficient for the total sum score was 0.832 ($p < 0.01$) [21]. Additionally, adherence to breathing exercises was assessed based on successful practice as well as maintaining the frequency of daily breathing exercises [22]. Patients were seen as adherence if they did the correct all steps through the practice checklist of breathing exercises and did one or more times per day within 10–15 minutes per time, and/or gradually increased by their own ability. Non-adherence was recorded for the patient if did not maintain daily practice or maintains daily practice but practiced with "fail" result. Finally, the patient was assessed as adherence to treatment if there was concurrent adherence with inhaled drugs and breathing exercise therapy; conversely, non-adherent if adherence one of two contents or non-adherence with both.

The patient's health status was assessed through two indicators, including severity of disease and degree of airway obstruction. For the first indicator, it was assessed through the modified Medical Research Council (mMRC) scale and the COPD Assessment Test (CAT) [23,24]. Patients were considered "Mild disease" if mMRC 0–1 and CAT < 10; and severity with mMRC = 2 and CAT > 10. The mMRC scale was tested with had good validity and reliability [25], while CAT scale was assessed with Cronbach's alpha coefficient of 0.924 [26]. The second indicator was a degree of airway obstruction that was measured by a spirometer, and the result of each patient was classified as:"Mild - Moderate" with FEV1 ≥ 80% and 50% < FEV1 < 80%); and "Severe" with FEV1 < 50%, respectively.

## Risk of bias

Our study may encounter some biases, particularly in the assessment of adherence, which is based on self-reporting. This method is vulnerable to desirability bias and recall bias, meaning that patients might over-report their adherence, especially knowing they are being monitored. To mitigate this risk, the study employed short, easy-to-understand questions, and researchers received training in data collection beforehand, which should help reduce bias to some extent.

Additionally, we combined self-reported measurements of medication adherence and breathing practices with direct observations of patients practicing their breathing exercises according to a checklist. Researchers conducted weekly assessments via phone video calls, monitored medication use diaries, and asked patients to send videos of their breathing exercises at home.

Another potential issue is dropout bias due to the 12-week intervention period, which may lead to a decrease in the number of study participants. To address this, we regularly monitored intervention activities on a weekly basis and thoroughly explained the research objectives and benefits of the intervention to the participants, which may help reduce dropout rates.

## Statistical analysis

Data analysis was performed using the Statistical Package for Social Sciences (SPSS, version 23.0). Testing for normal distribution with severity and dyspnea, the Kolmogorov-Smirnov test results have a significance level (Sig.) greater than

0.05. Continuous variables were calculated as means and variance, while regression analysis was used for examining the relationship between two variables. The difference statistically significant with p < 0.05.

### Ethical considerations

The study was approved by the Protocol Approval Council and the Medical Ethics Council of Nam Dinh University of Nursing, Decision No. 1681/GCN-HDDD dated August 2, 2021. All research participants were provided a detailed explaination about the purpose and purpose of the study by resreach team. All patients agreed to participate in the study voluntarily by signing a consent form that provided by the researcher at the first-time meeting. Participants were informed that they have the right to withdraw from the study at any time when they feel uncomfortable or unable to continue the study, and this experiment does not involve any form of inducement or coercion. Also, study does not carry risks that must compensate in money or in kind. After completing the intervention study, all participants in the control group were given a health consultation session and distributed documents with the same content as that in the intervention group.

All patients in the intervention group were clearly informed that participation in the intervention, including follow-up calls, was entirely voluntary. They had the option to decline or withdraw from the study at any time without any impact on their care or treatment. During follow-up calls, patients were reminded that attendance was not mandatory. Those who chose not to participate were recorded, and their decision was fully respected. No further follow-up calls were made to individuals who expressed their wish to discontinue.

## Results

### Participant's characteristics

After a 3-month intervention period, a total of 45 patients participated in the intervention group, and 45 patients participated in the control group. No patient gave up during the intervention; 12 group calls were conducted with 100% patient participation; 100% of patients participated in regular follow-up examinations once a month. According to the survey after the end of the intervention program, 100% of the subjects in our study were satisfied with the content, methods, and communication organization.

At baseline, the Chi-square test indicated no significant differences between the intervention and control groups in terms of treatment adherence, disease severity (measured by the CAT scale), and dyspnea (assessed using the mMRC scale) (p > 0.05) "Table 2".

**Table 2. Participant's characteristic.**

| Contents | Intervention group (n = 45) | | Control group (n = 45) | | P-value |
|---|---|---|---|---|---|
| | n | % | n | % | |
| **Treatment adherence**<br>Adherence<br>Non-adherence | 15<br>30 | 33.3<br>66.7 | 16<br>29 | 35.6<br>64.4 | p[a] > 0,05 |
| Severity of disease according to CAT scale (Mean;SD) | 14.38 ± 3.84 | | 14.31 ± 2.75 | | p[b] > 0,05 |
| Airway obstruction level according to mMRC scale (Mean;SD) | 2.27 ± 0.80 | | 2.21 ± 0.65 | | p[b] > 0,05 |

CAT, COPD Assessment Test; mMRC, modified Medical Research Council; SD, standard deviation

[a]p: p-value result of the analysis of two variables comparing two proportions using Chi-square test;

[b]p: The p-value result of the analysis of the difference in mean scores between two groups of values using Independent-Samples T-Test.

## Changing of treatment adherence of COPD outpatients after intervention

Table 3 highlights that prior to the intervention, adherence to inhaled medication was relatively low at 35.6%. However, after three months of intervention, this adherence rate significantly increased to 88.9%. Patients in the intervention group were 3.2 times more likely to adhere to inhaled medication compared to those in the control group, a result that was statistically significant (p = 0.002; OR = 3.2; 95% CI: 1.8–5.3). Similarly, initial adherence to breathing exercises was modest at 44.4%, but after three months, it rose to 84.4%. The intervention group demonstrated a 2.5 times higher adherence rate than the control group, with statistical significance (p = 0.000; OR = 2.5; 95% CI: 1.8–3.4). In terms of overall adherence, the intervention group exhibited a 2.2 times higher rate than the control group, which was also statistically significant (p = 0.001; OR = 2.2; 95% CI: 1.7–3.1) "Table 3".

## Changing of health status of COPD outpatients after intervention

The results indicated a significant improvement in disease severity (CAT ≥ 10 and mMRC ≥ 2) in the intervention group, with an odds ratio (OR) of 1.3 (95% CI: 0.8–1.7, p = 0.006). Likewise, the intervention group showed a notable improvement in the severity of airway obstruction among patients in the severe category, with an OR of 1.71 (95% CI: 0.74–3.95, p = 0.02) "Table 4".

## Discussion

Assessing treatment adherence of COPD patients is a challenge in clinical assessment and even in clinical research. Many studies have shown that adherence to long-term chronic disease treatment is suboptimal in real-world settings, and the WHO estimates that only 50% of patients adhere to their treatment regimens [27].

Table 3 illustrates that the adherence rate to treatment among participants was low, with only 35.6% in both groups at baseline. Our intervention took place when the Covid-19 epidemic was under control in Vietnam. Prior to this, from 2020 to 2022, ongoing outbreaks led large hospitals to prioritize resources for admitting and treating Covid-19 patients, which in turn diverted the management of COPD patients to local medical centers. This shift in treatment and care contributed to the low adherence rates. Many patients were hesitant to seek medical care due to the fear of transmitting the virus and the difficulties of traveling during social distancing measures. Additionally, some patients voluntarily discontinued their medications once they began feeling better, while others simply forgot to take their medication or ran out without the

**Table 3. Between-group comparisons respecting medication adherence status.**

| Group Outcomes | | Control group | | Intervenion group | | OR (95%CI) | P-value |
|---|---|---|---|---|---|---|---|
| | | Pretest (n = 45) | Posttest (n = 45) | Pretest (n = 45) | Posttest (n = 45) | | |
| Adherence to inhaled medications | Adherence | 16 (35.6) | 21 (46.7) | 15 (33.3) | 40 (88.9) | 3.2 (1.8 - 5.3) | 0.002 |
| | Non-adherence | 29 (64.4) | 24 (53.3) | 30 (66.7) | 5 (11.1) | | |
| Adherence to breathing exercises | Adherence | 20 (44.4) | 23 (51.1) | 15 (33.3) | 38 (84.4) | 2.5 (1.8 - 3.4) | <0.001 |
| | Non-adherence | 25 (55.6) | 22 (48.9) | 30 (66.7) | 7 (15.6) | | |
| Treatment adherence | Adherence | 16 (35.6) | 22 (48.9) | 16 (35.6) | 38 (84.4) | 2.2 (1.7 - 3.1) | 0.001 |
| | Non-adherence | 29 (64.4) | 23 (51.1) | 29 (64.4) | 7 (15.6) | | |

CAT, COPD Assessment Test; mMRC, modified Medical Research Council; OR, Odd ratio; CI, confidence interval.

**Table 4. Comparing rate of improving health status at baseline and after the intervention.**

| Group Outcomes | | Control group | | Intervenion group | | OR (95%CI) | P – value |
|---|---|---|---|---|---|---|---|
| | | Pretest (n = 45) | Posttest (n = 45) | Pretest (n = 45) | Posttest (n = 45) | | |
| Severity of disease | CAT < 10 and mMRC < 2 | 0 (0.0) | 0 (0.0) | 0 (0.0) | 8 (17.8) | | |
| | CAT ≥ 10 and mMRC ≥ 2 | 45 (100) | 45 (100) | 45 (100) | 37 (82.2) | 1.3 (0.8 - 1.7) | 0.006 |
| Degree of airway obstruction | Mild to Moderate | 33 (73.3) | 30 (66.7) | 29 (64.4) | 30 (66.7) | | |
| | Severe | 12 (26.7) | 15 (33.3) | 14 (31.2) | 10 (22.2) | 1.71 (0.74–3.96) | 0.02 |

OR, Odd ratio; CI, confidence interval.

means to purchase or obtain more. Therefore, implementing an educational intervention program at this time was seen as an ideal opportunity to address knowledge gaps, provide motivation, and improve patients' intentions regarding their treatment.

According to Emilsson et al. (2011), the concept of adherence emphasizes the need for consensus and can be defined as the extent to which a patient's behavior is consistent with the agreed recommendations from the prescriber and the medication [28]. To achieve optimal adherence, a patient must go through a series of key steps (from initiation, implementation to persistence or discontinuation of medication), and each step can be influenced by different factors [6,28]. Therefore, adherence is a key factor in the management of chronic respiratory diseases, especially COPD. In our study, a health education intervention conducted by nurses helped increase the treatment adherence rate of COPD outpatients by 84.4% compared to the control group at 48.9%. The treatment adherence rate in the intervention group was 2.2 times higher than in the control group, a statistically significant finding (p = 0.001; OR = 2.2; 95% CI: 1.7–3.1). The results of our study were higher than the increase in treatment adherence in previously conducted interventions [8,13].

Inhaled medications play an important role in the treatment of COPD patients [29]. This method of application has the advantage of delivering the drug directly into the airways, allowing for high local concentrations of the drug with a lower risk of systemic side effects [29]. Our study indicaed that at baseline, the rate of participants in the intervention and the control groups adherenting to inhaled medication was low (33.3% vs. 35.6%). After three months of intervention, adherence to inhaled medication use significantly increased to 88.9%. The adherence rate in the intervention group was 3.2 times higher than in the control group, a statistically significant result (p = 0.002; OR = 3.2; 95% CI: 1.8–5.3). To achieve the best treatment effect, patients must adhere to the prescribed regimen, use the inhaler correctly and with the appropriate inhalation technique [30]. While frequent inhaler use and correct technique are related, they are distinct concepts influenced by different factors [26]. This result was similar to the educational interventions of Khadela et al. (2020) [11], Ibrahim and El-Maksoud (2021) [12], and Tu et al (2019) [13]. But the study by Tu et al. (2019) showed that the intervention implemented by pharmacists with limited hospital pharmacist resources and conducted only in hospitals did not really bring much benefit to patients [13]. Otherwise, our intervention conducted by nurses brought a different view, specifically, nurses provided personal approach to patients, direct instruction and modeling, time for patients to repeat, and weekly meetings to answer questions together with reminding patients to use inhalers regularly and correctly at home. Therefore, maintaining continuous educational interventions by community nurses really improves medication adherence for COPD patients in the community.

Performing breathing exercises is an inexpensive and easy-to-perform physical therapy for COPD patients and is considered an important component of pulmonary rehabilitation [31]. Regarding adherence to breathing exercise, our

study found that the rate of adherence in both groups was low at the begining. After the intervention, this rate increased significantly by 84.4% for the intervention group, and the rate of compliance with breathing exercises was 2.5 times higher in the intervention group than in the control group, and this result was statistically significant (p = 0.000 with OR = 2.5; 95%CI = 1.8–3.4). This result was similar to the study of Li et al. (2022) in China [32], and the study of Rossi et al. (2014) in Brazil [33]. Accordingly, both authors described that pursed lip practice can contribute to prolonging the exhalation time, promoting increased external airway resistance, to help expel residual air in the lungs, while diaphragmatic breathing can increase the efficiency of diaphragm contraction, thereby preventing the occurrence of opposite inspiratory movements of the chest and abdomen. The combination of these two methods has a synergistic effect and promotes the improvement of lung function in patients with COPD. Therefore, the implementation of educational intervention through the guidance and modeling of nurses to help patients perform these two breathing exercises with regular frequency and in the correct step-by-step manner at home has brought about very positive results.

On the other hand, treatment adherence was assessed simultaneously based on two measures, including adherence to inhaler medication and adherence to breathing exercises. This was a new way to assess treatment adherence for COPD patients and not found in any previous research. In this intervention program, we organized two direct educational consultation sessions at the outpatient clinic, including a session promoting knowledge of COPD, belief in medication, and disease self-management measures; and another session focusing on instructions on how to use an inhaler's drug, breathing exercises therapy, as well as providing video links for patients to watch again at home. In addition, we implemented weekly group phone calls to support patients at home. Conducting health education and ongoing patient support has shown a positive impact of the intervention program on treatment adherence in COPD patients. This positive outcome may be explained by basing our intervention program based on the assessment of experts and the patients themselves, ensuring that the program met criteria related to acceptability, appropriateness, feasibility, and the ability to access and meet gaps in knowledge and skills for COPD patients [22].

Prevention of exacerbations and death due to exacerbations is the main goal of COPD treatment, and it depends on improving medication adherence [3]. Before the intervention, the treatment adherence rate of all participants in both groups were low. After the intervention, the treatment adherence rate of the intervention group increased very high, but the control group maintained a low rate; the difference between the groups, before and after the intervention, was statistically significant (p < 0.05). Our study showed the importance of conducting an intervention program to improve treatment adherence behavior based on the Theory of Planned Behavior; this result was similar to previous studies that was implemented with chronically ill patients [34–36].

The results of the severity of disease of the COPD outpatient intervention group after three months also showed that the patients' scores of mMRC < 2 and CAT < 10 increased from 0.0% to 17.8%, the results demonstrated a significant improvement in disease severity (CAT ≥ 10 and mMRC ≥ 2) in the intervention group, with an odds ratio (OR) of 1.3 (95% CI: 0.8–1.7, p = 0.006).and our results were similar to other studies [8,13]. This showed the effectiveness of the educational intervention program to promote treatment adherence for outpatients and once again confirms the effectiveness of monitoring and predicting COPD status through the CAT and mMRC scales. Therefore, at outpatient clinics, healthcare staff need to increase the use of CAT and mMRC scales to assess disease stability over time, especially in units that do not have the conditions to measure ventilation function. In addition, staff also need to provide specific instructions to patients and their families about the role, meaning, and usefulness of CAT and the mMRC scale so that they can self-assess the severity of COPD at home, thereby raising patients' awareness of the disease's management.

COPD is manifested by airway obstruction, which tends to gradually worsen and irreversible. Without an intervention the patient's ventilation function rapidly decreases [7]. Results before and after three months of intervention show that the patient's level of congestion has changed in a positive direction. The severity of airway obstruction in the severe group showed a significant improvement in the intervention group, with an odds ratio (OR) of 1.71 (95% CI: 0.74–3.95, p = 0.02). This result was similar to previous studies [8,34]. The ventilatory function of people with COPD often gradually decreases

over time, and the degree of rapid or slow decline depends on the elimination of risk factors, medication adherence, and adherence with respiratory rehabilitation. Our research demonstrates that simple health education counseling measures such as instructing patients to comply with medication and performing breathing exercises would have the effect of slowing down the development process of COPD, thereby helping to improve the patient's quality of life.

### Limitation

First, the study was conducted during the Covid-19 pandemic, which had only recently been brought under control. As a result, patients were still apprehensive and concerned about the risk of spreading the virus when visiting the hospital. This made patient recruitment challenging, and organizing intervention sessions posed its own difficulties. Additionally, working with a large group of patients raised clinical concerns due to proximity issues. To address this, we received active support from two hospitals to better divide the patient groups. All participants, including their family members, were tested for Covid-19 each time they visited the clinic. The counseling and explanations provided to the participants were well received, ensuring that everyone felt secure in engaging with the entire intervention program. Second, the intervention was implemented over a relatively short duration of three months, which raises the possibility that the results obtained may be temporary and not sustainable in the long term. Future studies should prioritize longitudinal research at the six- and twelve-month marks, while also incorporating additional reinforcement strategies. This approach could yield valuable insights into sustaining improved adherence to COPD management practices over time, aligning more closely with the GOLD 2024 recommendations for comprehensive and sustainable patient-centered care. Third, this study focused on patients with stable health conditions who were receiving outpatient treatment. Future research should consider extending interventions to severely ill patients undergoing inpatient care, which would allow for necessary adjustments to enhance the intervention's effectiveness and applicability across a broader range of patient populations. Fourth, the study involved a sample of individuals with stable COPD, limiting the generalizability of the results to all COPD patients. Additionally, the study received approval from the ethics committees at the university and the two hospitals prior to patient implementation. However, the research team faced challenges in understanding the international registration process for randomized control trials, resulting in the registration code being obtained only after data collection was complete. The research team remained committed to adhering to ethical regulations throughout the implementation process.

### Conclusions

Intervention activities significantly improved treatment adherence of outpatients with COPD. The control group initially evaluated the adherence rate to be 35.6%; after 3 months, this rate increased to only 48.9%; meanwhile, the rate in the pre-intervention group was 35.6%, and after the intervention, the rate increased significantly to 84.4%. The patient's heath status showed a decrease in severity of disease and airway obstruction for the group receiving the education intervention. From the effective of intervention program, health care professionals (especially clinical nurses) should consider implementing a broader program to help people with COPD proactively manage and improve their health in a conscious way. Future research should prioritize longitudinal studies at six- and twelve-months incorporating reinforcement strategies consistent with the GOLD 2024 guidelines, extending interventions to critically ill inpatients to improve intervention efficacy and applicability to a larger population.

### Supporting information

**S1 File. The CONSORT checklist.**
(PDF)

**S2 File. Study protocol.**
(PDF)

## Acknowledgments

We would like to express our sincere thanks to the study participants for their enthusiastic cooperation; and medical staffs of two outpatient clinic of C – Da Nang hospital, and Da Nang hospital for lung disease who supported us effectively during the time of the study performing.

## Author contributions

**Conceptualization:** Tran Huu Thong, Nguyen Thi Thu Trieu.

**Data curation:** Tran Huu Thong, Nguyen Thi Thu Trieu, Nguyen Thi Yen Hoai, Nguyen Thi Thu Hien.

**Formal analysis:** Tran Huu Thong, Nguyen Thi Thu Trieu, Nguyen Thi Yen Hoai, Nguyen Thi Thu Hien.

**Investigation:** Tran Huu Thong, Nguyen Thi Thu Trieu, Nguyen Thi Thu Hien.

**Methodology:** Tran Huu Thong, Nguyen Thi Thu Trieu, Nguyen Thi Yen Hoai, Nguyen Thi Thu Hien.

**Project administration:** Tran Huu Thong, Nguyen Thi Thu Trieu.

**Resources:** Tran Huu Thong, Nguyen Thi Thu Trieu, Nguyen Thi Yen Hoai, Nguyen Thi Thu Hien.

**Software:** Tran Huu Thong, Nguyen Thi Thu Trieu, Nguyen Thi Thu Hien.

**Supervision:** Nguyen Thi Thu Hien.

**Validation:** Tran Huu Thong, Nguyen Thi Thu Trieu, Nguyen Thi Yen Hoai, Nguyen Thi Thu Hien.

**Visualization:** Tran Huu Thong, Nguyen Thi Thu Trieu, Nguyen Thi Yen Hoai, Nguyen Thi Thu Hien.

**Writing – original draft:** Tran Huu Thong, Nguyen Thi Thu Trieu, Nguyen Thi Yen Hoai, Nguyen Ngoc Tung, Nguyen Thi Thu Hien.

**Writing – review & editing:** Tran Huu Thong, Nguyen Thi Thu Trieu, Nguyen Thi Yen Hoai, Nguyen Ngoc Tung, Nguyen Thi Thu Hien.

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
