## [Decision Letter · Decision Letter 0]

PONE-D-25-08339Effect of Health Education Intervention on Treatment Adherence and Health Status in Patients with Chronic Obstruction Pulmonary Disease: A Random Control TrialPLOS ONE

Dear Dr. Thong,

Thank you for submitting your manuscript to PLOS ONE. After careful consideration, we feel that it has merit but does not fully meet PLOS ONE’s publication criteria as it currently stands. Therefore, we invite you to submit a revised version of the manuscript that addresses the points raised during the review process.

**ACADEMIC EDITOR: This study has the potential to provide valuable evidence on health education interventions and outcomes for COPD in Vietnam, an area that is currently lacking research. The topic is of interest. However, please ensure that all comments from the reviewers are addressed, particularly the methodology section, which needs further refinement.**

We look forward to receiving your revised manuscript.

Kind regards,

Thien Tan Tri Tai Truyen, M.D.

Academic Editor

PLOS ONE

5. Please include a caption for figure 1.

Reviewers' comments:

Reviewer's Responses to Questions

**Comments to the Author**

1. Is the manuscript technically sound, and do the data support the conclusions?

Reviewer #1: Partly

Reviewer #2: Yes

2. Has the statistical analysis been performed appropriately and rigorously? 

Reviewer #1: No

Reviewer #2: Yes

3. Have the authors made all data underlying the findings in their manuscript fully available?

Reviewer #1: Yes

Reviewer #2: Yes

4. Is the manuscript presented in an intelligible fashion and written in standard English?

Reviewer #1: No

Reviewer #2: Yes

5. Review Comments to the Author

Reviewer #1: COPD is a major global health burden, especially in low- and middle-income countries. Your study addresses a crucial aspect of COPD management—treatment adherence—by evaluating a health education intervention.

Following are my comments:

1. Methodological Concerns

A. Selection of Control Group

The control group was selected from a different hospital (Da Nang Lung Hospital), while the intervention group was from C-Da Nang Hospital.

Differences in hospital protocols, healthcare provider behavior, and patient populations could introduce confounding variables.

Ideally, both groups should be drawn from the same pool and randomized within the same hospital to minimize systematic differences.

B. Sample Size Justification

i. The calculation assumes an expected adherence increase from 50% to 85%, which is a very optimistic improvement.

This large effect size may not be realistic and could lead to underpowered results if actual improvements are smaller.

ii. High Attrition Rate Consideration

While the study factored in a 30% dropout rate, 100% of patients completed the intervention, which is highly unusual for behavioral studies.

A discussion on how dropout bias was avoided would strengthen credibility.

C. Intervention Implementation and Measurement Bias

i. Self-Reported Adherence Measurements

The adherence assessment (TAI-10 and breathing exercise checklist) is based on self-reporting, which is prone to social desirability bias and recall bias.

Patients may over-report adherence, especially when aware of being monitored.

Consider adding an objective measure, such as electronic monitoring devices for inhalers or video-confirmed breathing exercises.

ii. Blinding of Outcome Assessors

The research team who provided the intervention also assessed adherence outcomes, which may introduce observer bias.

Ideally, assessors should be blinded to the intervention group to minimize bias in adherence reporting.

iii. Lack of Long-Term Follow-Up

The study only measures adherence and health status at three months post-intervention.

A longer follow-up (e.g., 6 or 12 months) would provide insights into whether adherence improvements are sustained over time.

2. Statistical and Data Reporting Issues

A. No Effect Size Reporting

While statistical significance (p-values) is provided, effect sizes (e.g., Cohen’s d for t-tests, odds ratios for categorical outcomes) should be included to quantify the magnitude of changes.

Reporting confidence intervals (CIs) would improve result interpretation.

B. Comparison of Airway Obstruction (FEV1 Stages)

The study claims a positive shift in airway obstruction stages, but no statistical significance (p > 0.05) is reported.

This suggests the improvement could be due to chance rather than the intervention.

More detailed subgroup analysis is needed to explore trends in different severity groups.

3. Literature Review and Framing Issues

A. Limited Discussion of Existing Multicomponent Interventions

The introduction highlights previous studies focusing on either medication adherence or breathing exercises but does not discuss combined interventions.

Are there prior integrated adherence interventions for COPD that your study builds upon? A more thorough review would strengthen justification.

B. GOLD 2024 Citation Needs More Context

The study references GOLD 2024 but does not elaborate on how the intervention aligns with GOLD-recommended strategies for adherence improvement.

Explicitly linking your intervention elements to GOLD recommendations would add clarity.

C. More Critical Analysis of Vietnam-Specific Barriers

The study emphasizes Vietnam's limited research on COPD adherence but does not explore country-specific barriers (e.g., healthcare accessibility, cultural beliefs, financial constraints).

A deeper discussion on why adherence is particularly low in Vietnam would provide better context.

4. Ethical and Practical Considerations

A. Potential Coercion in Phone Call Participation

Patients were called three times if they missed a session, which may feel coercive.

Clarify if patients were free to decline participation without consequences.

B. Lack of Standardization in Group Calls

Patients were reminded about adherence via group calls and personal calls, but individual responses may have been influenced by peer pressure.

Were all group calls structured identically? Standardizing these interactions would enhance consistency.

Clarity & Conciseness

5. Discussion

The discussion is lengthy and repetitive in certain areas, particularly in emphasizing adherence statistics multiple times. Condensing these sections can enhance readability.

Some sentences are complex and need restructuring for better flow. Example:

“In fact, adherenting inhaler frequent and technique are closely related but quite independent concepts and can be influenced by different factors [26].”

Suggestion: “While frequent inhaler use and correct technique are related, they are distinct concepts influenced by different factors [26].”

6. Scientific Rigor & Interpretation of Results

A. The interpretation of airway obstruction improvement could be overstated, as the difference in obstruction severity (stage 3 to stage 2) was not statistically significant (p > 0.05).

Consider rewording to acknowledge that while a positive trend was observed, the results do not conclusively prove a long-term change in lung function.

B. Short-Term Nature of Study

The discussion acknowledges the 3-month limitation, but a more detailed recommendation for longitudinal follow-up would strengthen the argument.

Suggestion: Consider whether behavioral adherence improvements persist beyond 3 months or if patients require continued reinforcement.

C. Generalisability Issues

The discussion correctly states that the findings apply only to stable COPD patients. However, a more nuanced discussion on how this intervention might work in severe COPD or hospitalized patients would be valuable.

Your study is well-structured, methodologically sound, and addresses a critical issue in COPD management. However, some methodological concerns—selection bias, self-reported adherence, lack of long-term follow-up, and absence of blinding—should be addressed to enhance reliability.

Reviewer #2: The present is an interesting paper. Some issues should be added

1) primary end point should be clearly stated in the abstract

2)probably for a typo abstract is double reported

3) sample size in the methods>a reference should be added for 50%

4) it should be added if randomization was performed through block and how many patients

5)normal distribution should be checked for

6) due to the reduced sample size, do authors think that multivariate analysis may be of help

6. PLOS authors have the option to publish the peer review history of their article (what does this mean? ). If published, this will include your full peer review and any attached files.

**Do you want your identity to be public for this peer review?** For information about this choice, including consent withdrawal, please see our Privacy Policy .

Reviewer #1: No

Reviewer #2: **Yes: ** Fabrizio D'Ascenzo

---

## [Author Response · Author response to Decision Letter 1]

31 Mar 2025

Dear Editor,

Thank you for the suggestions to improve our manuscript titled, “Effect of Health Education Intervention on Treatment Adherence and Health Status in Patients with Chronic Obstruction Pulmonary Disease: A Random Control Trial”. Please see the following table listing requested revisions and our response.

A. Journal requirements

1. Please ensure that your manuscript meets PLOS ONE's style requirements, including those for file naming

- Response: Done

- Response: We added the content of patients signed the consent form in Ethical consideration part (Method)

3. We note that you have indicated that there are restrictions to data sharing for this study. PLOS only allows data to be available upon request if there are legal or ethical restrictions on sharing data publicly.

- Response: Done

- Response: Done

5. Please include a caption for figure 1.

- Response: Done

- Response: Done

B. 1st Reviewers' comments

1. Methodological Concerns

1a. Selection of Control Group

The control group was selected from a different hospital (Da Nang Lung Hospital), while the intervention group was from C-Da Nang Hospital.

Differences in hospital protocols, healthcare provider behavior, and patient populations could introduce confounding variables.

Ideally, both groups should be drawn from the same pool and randomized within the same hospital to minimize systematic differences.

- Response: We added contents about the similarities between the two study sites, specifically in section 2.2 of Methods parts. Besides, about the similarities between two groups, we added a content to make it clearly.

1b. Sample Size Justification

i. The calculation assumes an expected adherence increase from 50% to 85%, which is a very optimistic improvement.

This large effect size may not be realistic and could lead to underpowered results if actual improvements are smaller.

- Response: We added reference [13] that was a previous study improving post-intervention of nearly 20%.

ii. High Attrition Rate Consideration

While the study factored in a 30% dropout rate, 100% of patients completed the intervention, which is highly unusual for behavioral studies.

A discussion on how dropout bias was avoided would strengthen credibility.

- Response: 100% of patients completed the intervention in our study could be explained below:

The study combined various measures such as: consulting, direct practice instructions, providing videos for patients to practice at home, weekly monitoring by phone, monitoring through the study subject's diary and periodic re-examination. The reason this model is effective is because it meets all the criteria: (1) is suitable and feasible; (2) is sustainable; (3) is accessible, meeting the shortage for patients. In particular, the intervention was carried out in the context of the COVID-19 epidemic having just been controlled in Da Nang city. It can be said that the outbreaks have been continuous in Da Nang city since the beginning of 2020, especially in 2021 to mid-2022, major hospitals in Da Nang city have focused resources on admitting and treating COVID-19 patients, and the management of people with COPD is transferred to commune/ward health facilities. At that time, COPD patients had limited access to care support and were almost disconnected from their primary treatment facility. Therefore, when the epidemic was under control, the educational intervention program at the time of the study was considered a golden time to supplement the knowledge gap, improve practical skills for patients when they returned to their old treatment facility.

1c. Intervention Implementation and Measurement Bias

i) Self-Reported Adherence Measurements

The adherence assessment (TAI-10 and breathing exercise checklist) is based on self-reporting, which is prone to social desirability bias and recall bias.

Patients may over-report adherence, especially when aware of being monitored.

Consider adding an objective measure, such as electronic monitoring devices for inhalers or video-confirmed breathing exercises.

- Response: We added a 2.7 section to explain the risk of bias of this study and actions we did mo minimize those bias.

ii. Blinding of Outcome Assessors

The research team who provided the intervention also assessed adherence outcomes, which may introduce observer bias.

Ideally, assessors should be blinded to the intervention group to minimize bias in adherence reporting.

- Response: We added the content in section 2.3

iii. Lack of Long-Term Follow-Up

The study only measures adherence and health status at three months post-intervention.

A longer follow-up (e.g., 6 or 12 months) would provide insights into whether adherence improvements are sustained over time.

- Response: We added this content in Limitation section.

2. Statistical and Data Reporting Issue

2a. No Effect Size Reporting

While statistical significance (p-values) is provided, effect sizes (e.g., Cohen’s d for t-tests, odds ratios for categorical outcomes) should be included to quantify the magnitude of changes.

Reporting confidence intervals (CIs) would improve result interpretation.

- Response: We added this content in Table 3, and rewrited the result in Result part (section 3.2)

2b. Comparison of Airway Obstruction (FEV1 Stages)

The study claims a positive shift in airway obstruction stages, but no statistical significance (p > 0.05) is reported.

This suggests the improvement could be due to chance rather than the intervention.

More detailed subgroup analysis is needed to explore trends in different severity groups.

- Response: We re-analysed and shown in Table 4, rewrited the result in Result part (section 3.3)

3. Literature Review and Framing Issues

3a. Limited Discussion of Existing Multicomponent Interventions

The introduction highlights previous studies focusing on either medication adherence or breathing exercises but does not discuss combined interventions.

Are there prior integrated adherence interventions for COPD that your study builds upon? A more thorough review would strengthen justification.

- Response: We added more content as reviewer suggested in Introduction part

3b. GOLD 2024 Citation Needs More Context

The study references GOLD 2024 but does not elaborate on how the intervention aligns with GOLD-recommended strategies for adherence improvement.

Explicitly linking your intervention elements to GOLD recommendations would add clarity.

- Response: We added the GOLD content in Introduction part, paragraph 5

3c. More Critical Analysis of Vietnam-Specific Barriers

The study emphasizes Vietnam's limited research on COPD adherence but does not explore country-specific barriers (e.g., healthcare accessibility, cultural beliefs, financial constraints).

A deeper discussion on why adherence is particularly low in Vietnam would provide better context.

- Response: We added this content in Disscussion part, paragraph 2.

4. Ethical and Practical Considerations

4a. Potential Coercion in Phone Call Participation

Patients were called three times if they missed a session, which may feel coercive.

Clarify if patients were free to decline participation without consequences.

- Response: We added this explain in Ethical consideration part.

4b. Lack of Standardization in Group Calls

Patients were reminded about adherence via group calls and personal calls, but individual responses may have been influenced by peer pressure.

Were all group calls structured identically? Standardizing these interactions would enhance consistency.

- Response: We explained this content in section 2.4 (end of paragraph 2) of Method part

5. Discussion

The discussion is lengthy and repetitive in certain areas, particularly in emphasizing adherence statistics multiple times. Condensing these sections can enhance readability.

Some sentences are complex and need restructuring for better flow. Example:

“In fact, adherenting inhaler frequent and technique are closely related but quite independent concepts and can be influenced by different factors [26].”

Suggestion: “While frequent inhaler use and correct technique are related, they are distinct concepts influenced by different factors [26].”

- Response: We recheck and fixed those problem in Discussion and we rewrite the sentence as reviewer suggested.

6. Scientific Rigor & Interpretation of Results

6a. The interpretation of airway obstruction improvement could be overstated, as the difference in obstruction severity (stage 3 to stage 2) was not statistically significant (p > 0.05).

Consider rewording to acknowledge that while a positive trend was observed, the results do not conclusively prove a long-term change in lung function.

- Response: We refixed the content as reviewer suggestion on Table 4

6b. Short-Term Nature of Study

The discussion acknowledges the 3-month limitation, but a more detailed recommendation for longitudinal follow-up would strengthen the argument.

Suggestion: Consider whether behavioral adherence improvements persist beyond 3 months or if patients require continued reinforcement.

- Response: We added this problem in the Limitation part

6c. Generalisability Issues

The discussion correctly states that the findings apply only to stable COPD patients. However, a more nuanced discussion on how this intervention might work in severe COPD or hospitalized patients would be valuable.

Your study is well-structured, methodologically sound, and addresses a critical issue in COPD management. However, some methodological concerns—selection bias, self-reported adherence, lack of long-term follow-up, and absence of blinding—should be addressed to enhance reliability.

- Response: We added this problem in the Limitation part

C. 2nd Reviewers' comments

Some issues should be added

1) primary end point should be clearly stated in the abstract

- Response: We added the primary result in Abstract

2) probably for a typo abstract is double reported

- Response: We fixed it

3) sample size in the methods>a reference should be added for 50%

- Response: We have already added reference for this content

4) it should be added if randomization was performed through block and how many patients

- Response: We added this in Randomization and blinding part (section 2.3 of Method)

5) normal distribution should be checked for

- Response: We added this in Data analysed part

6) due to the reduced sample size, do authors think that multivariate analysis may be of help

- Response: We agree that multivariate analysis can be a valuable tool to account for confounding factors and explore relationships between variables in a more robust manner. However, the reduced sample size in our study poses certain limitations, including:

Statistical Power:

Multivariate analysis often requires a sufficiently large sample size to ensure reliable and accurate results. With the reduced sample size in our study, there is a risk of overfitting the model or drawing conclusions that may lack statistical robustness.

Alternative Approaches: To address the issue of confounding factors, we opted for simpler statistical methods, such as univariate analysis, to minimize the risk of overfitting while still providing meaningful insights.

---

## [Decision Letter · Decision Letter 1]

Effect of Health Education Intervention on Treatment Adherence and Health Status in Patients with Chronic Obstruction Pulmonary Disease: A Random Control Trial

PONE-D-25-08339R1

Dear Dr. Thong,

We’re pleased to inform you that your manuscript has been judged scientifically suitable for publication and will be formally accepted for publication once it meets all outstanding technical requirements.

Kind regards,

Thien Tan Tri Tai Truyen, M.D.

Academic Editor

PLOS ONE

Additional Editor Comments (optional):

Reviewers' comments:

Reviewer's Responses to Questions

**Comments to the Author**

1. If the authors have adequately addressed your comments raised in a previous round of review and you feel that this manuscript is now acceptable for publication, you may indicate that here to bypass the “Comments to the Author” section, enter your conflict of interest statement in the “Confidential to Editor” section, and submit your "Accept" recommendation.

Reviewer #1: All comments have been addressed

Reviewer #2: All comments have been addressed

Reviewer #3: All comments have been addressed

Reviewer #4: All comments have been addressed

2. Is the manuscript technically sound, and do the data support the conclusions?

Reviewer #1: (No Response)

Reviewer #2: (No Response)

Reviewer #3: Yes

Reviewer #4: Yes

3. Has the statistical analysis been performed appropriately and rigorously? 

Reviewer #1: (No Response)

Reviewer #2: (No Response)

Reviewer #3: Yes

Reviewer #4: Yes

4. Have the authors made all data underlying the findings in their manuscript fully available?

Reviewer #1: (No Response)

Reviewer #2: (No Response)

Reviewer #3: Yes

Reviewer #4: Yes

5. Is the manuscript presented in an intelligible fashion and written in standard English?

Reviewer #1: (No Response)

Reviewer #2: (No Response)

Reviewer #3: Yes

Reviewer #4: Yes

6. Review Comments to the Author

Reviewer #1: (No Response)

Reviewer #2: (No Response)

Reviewer #3: I wasn't involved in the first round of peer-review, but I agree the authors did a good job addressing the reviewers' concerns, and provided explainations for limitations of the study. In all, I believe the manuscript is good for publication.

Reviewer #4: This is an interesting and relevant manuscript that aims to evaluate the effectiveness of the health education intervention with the purposes of evaluating the effectiveness of a health education intervention on treatment adherence behaviour of COPD outpatients

The research idea is fairly novel, and the methodology used make this study interesting and provides important conclusions with a potential impact in clinical practice and in health care planning and organization.

The study design is well structured and follows the construction recommended for a randomised control trial.

The discussion is well structured, pertinent, and clear and the conclusions are relevant and based on the results.

Some questions raised by previous reviewers were methodically answered by the authors in an accurate way.

7. PLOS authors have the option to publish the peer review history of their article (what does this mean? ). If published, this will include your full peer review and any attached files.

**Do you want your identity to be public for this peer review?** For information about this choice, including consent withdrawal, please see our Privacy Policy .

Reviewer #1: No

Reviewer #2: **Yes: ** Fabrizio D'Ascenzo

Reviewer #3: No

Reviewer #4: No

---

## [Editor Report · Acceptance letter]

PONE-D-25-08339R1

PLOS ONE

Dear Dr. Thong,

I'm pleased to inform you that your manuscript has been deemed suitable for publication in PLOS ONE. Congratulations! Your manuscript is now being handed over to our production team.

Kind regards,

on behalf of

Dr. Thien Tan Tri Tai Truyen

Academic Editor

PLOS ONE